# Ferroptosis: The Potential Target in Heart Failure with Preserved Ejection Fraction

**DOI:** 10.3390/cells11182842

**Published:** 2022-09-12

**Authors:** Qing Li, Zhiqiang Zhao, Xia Zhou, Yuting Yan, Lusi Shi, Jiafan Chen, Baohui Fu, Jingyuan Mao

**Affiliations:** 1Department of Cardiovascular Diseases, First Teaching Hospital of Tianjin University of Traditional Chinese Medicine, National Clinical Research Center for Chinese Medicine Acupuncture and Moxibustion, Tianjin 300381, China; 2Graduate School, Tianjin University of Traditional Chinese Medicine, Tianjin 301617, China

**Keywords:** ferroptosis, heart failure with preserved ejection fraction, iron overload, oxidative stress, calcium ion

## Abstract

Ferroptosis is a recently identified cell death characterized by an excessive accumulation of iron-dependent reactive oxygen species (ROS) and lipid peroxides. Intracellular iron overload can not only cause damage to macrophages, endothelial cells, and cardiomyocytes through responses such as lipid peroxidation, oxidative stress, and inflammation, but can also affect cardiomyocyte Ca^2+^ handling, impair excitation–contraction coupling, and play an important role in the pathological process of heart failure with preserved ejection fraction (HFpEF). However, the mechanisms through which ferroptosis initiates the development and progression of HFpEF have not been established. This review explains the possible correlations between HFpEF and ferroptosis and provides a reliable theoretical basis for future studies on its mechanism.

## 1. Introduction

Ferroptosis, a unique form of cell death that differs from necrosis and apoptosis, was first proposed in 2012 [1]. Ferroptosis is characterized by an excess of iron-dependent reactive oxygen species (ROS) and lipid peroxides. Iron is a key component of hemoglobin, myoglobin, and various oxidase and mitochondrial respiratory chain proteins involved in cellular metabolism [2] and is, therefore, essential for many biochemical and metabolic processes. Under conditions of iron overload, iron in plasma exceeds transferrin-binding capacity, and nontransferrin-bound iron (NTBI) appears in the bloodstream. Excessive absorption of NTBI results in the production of highly ROS, which causes peroxidation of membrane lipids and oxidative damage to intracellular proteins, ultimately leading to cell death. As one of the most metabolically active tissues, the myocardium is highly sensitive to iron overload.

Different from heart failure with reduced ejection fraction (HFrEF), heart failure with preserved ejection fraction (HFpEF) is a multisystemic syndrome, with most patients having a history of metabolic disorders such as obesity, diabetes, hypertension, and hyperlipidaemia [3,4]. Several studies have consistently shown that the presence of these metabolic comorbidities, which result in oxidative stress and chronic low-grade inflammatory states, is a main link to HFpEF. Increased levels of ROS frequently attack the cellular genome, proteins, and lipids, leading to a rise in lipid peroxidation (LPO) products, disruption of endogenous antioxidant mechanisms, and a decrease in glutathione (GSH) levels [5]. Furthermore, this systemic pro-inflammatory state can affect coronary microvascular function and compromise cardiomyocyte structure and cardiac function [6], implying a connection between inflammation, oxidative stress of ferroptosis, and HFpEF. In addition, studies have shown that left ventricular (LV) diastolic function may be a more sensitive early marker of myocardial iron overload than systolic function [7,8]. Iron overload and its dependence on LPO, oxidative stress, and inflammatory response may play an important role in the pathological process of HFpEF. Therefore, we summarize the biochemical regulation of ferroptosis to further discuss its relevance with HFpEF and its related risk factors and demonstrate progress in the study of anti-ferroptosis treatment for HFpEF, providing new ideas and targets for treatment.

## 2. Biochemical Regulation of Ferroptosis

The initiation and execution of ferroptosis is a highly regulated process. Although the specific mechanism of iron death has not yet been clarified, with the continuous expansion of research, several key ways to mediate ferroptosis have been identified, including LPO, amino acid metabolism, and iron metabolism (Figure 1).

### 2.1. Iron Overload

Iron is required for both ferroptosis execution and the accumulation of lipid peroxide. Extracellular Fe^3+^ is bound to transferrin and pumped into the cell via the transferrin receptor1 (TFR1) and subsequently reduced to Fe^2+^ by intracytoplasmic iron oxide reductase3 (STEAP3). Then, Fe^2+^ is released into a labile iron pool in the cytoplasm from the endosome mediated by divalent metal transporter protein 1 (DMT1). Excess Fe^2+^ can activate both the non-enzymatic [9] and enzymatic [10] pathways of LPO via the Fenton reaction and by catalyzing oxygenase activity to induce ferroptosis. Fe^2+^ is excreted from cells through the iron export protein ferroportin (FPN1) and then converted to Fe^3+^ with low redox activity by ceruloplasmin (CP). Iron homeostasis is regulated by iron import, transport, storage, and export processes. The addition of bioavailable forms of iron, such as ferric ammonium citrate, ferric citrate, ferric chloride hexahydrate, and iron bound to transferrin, sensitizes cells to develop ferroptosis [11]. On the contrary, iron chelators, such as deferoxamine and ciclopirox, can inhibit ferroptosis through iron starvation [12]. A study of ferritin H (Fth1) knockout mice fed a high-iron diet showed reduced levels of GSH and increased LPO, which were saved using Ferrostatin-1, a ferroptosis inhibitor [13].

### 2.2. Lipid Peroxidation

LPO triggered by ROS attack on membrane polyunsaturated fatty acids (PUFAs) is the direct cause of ferroptosis. LPO products can destabilize the phospholipid bilayer and eventually cause the disintegration of the cell membrane [14]. ROS hydroxyl radicals (OH^−^) and peroxynitrite (ONOO^−^) are two particularly chemically reactive species of activated oxygen that can induce LPO. The OH- formed by the Fenton reaction from intracellular free Fe^2+^ can build lipid peroxides directly with PUFAs in membrane phospholipids and attack the cytosolic membrane, causing iron death morphological changes [9]. The ONOO^−^ can be produced by nitrosative stress [15]. In addition, LPO can occur enzymatically by lipoxygenase (LOX), cyclooxygenase (COX), cytochrome P450 (CYP450), and myeloperoxidase (MPO). Inhibition of LOX has been shown to inhibit RSL3-induced ferroptosis [16]. Non-enzymatic LPO could play a key role in driving ferroptosis as it produces a mixture of non-specific stereoisomers that are highly cytotoxic [10,17]. Derivatives from the decomposition of lipid peroxides, including 4-hydroxynonenal (4-HNE) and malondialdehyde (MDA), can react with nucleic acids and proteins, leading to further cellular damage, and these derivatives can also be regarded as important molecular markers for the detection of ferroptosis and LPO [18,19,20].

### 2.3. Amino Acid Metabolism

Amino acid metabolism is involved in the regulation of ferroptosis by LPO. Both cystine [21] and glutamate [11] are vital molecules that regulate ferroptosis. Normally, the cystine/glutamate antiporter (system Xc^−^) in the cell membrane takes in one molecule of cystine and excretes one molecule of glutamate in a 1:1 ratio. Cystine enters the cell and then is reduced to cysteine, which is involved in the synthesis of glutathione (GSH). GPX4 (an important antioxidant enzyme) uses GSH as a substrate to scavenge lipid peroxidation and reduce oxidative stress. Drugs that reduce intracellular cysteine and GSH by targeting the system Xc^−^, such as erastin, or drugs that directly inhibit GPX4, such as RSL3, have been shown to induce ferroptosis [1,12,22]. Glutamine is also transported into the cell through the SLC1A5/SLC38A1 transporter and then converted to glutamate by glutaminase (GLS). Gao et al. have found that a GLS inhibitor, blocking glutamine glutamate production, is an alternative way to prevent ferroptosis [11]. Furthermore, a recent study has shown another potential mechanism of ferroptosis resistance, that is, FSP1 reduces the coenzyme Q10 (CoQ10) under the action of NADPH to prevent lipid peroxidation [23].

## 3. The Comorbidity–Inflammation Paradigm in HFpEF Is Closely Related to the Regulatory Mechanism of Ferroptosis

### 3.1. The Comorbidity–Inflammation Paradigm in HFpEF

Over the past decade, the understanding of HFpEF has progressed from a basic diastolic malfunction of the LV to a syndrome caused by multiple system diseases. Obesity, diabetes, hypertension, and hyperlipidemia are all diseases that can cause and keep the body in a chronic inflammatory state, which is considered to be the initial event of diastolic dysfunction in HFpEF. There was a significant increase in plasma levels of CRP in patients with HFpEF with an increasing number of comorbidities [24]. In addition to CRP, inflammatory biomarkers such as IL-1β, IL-6, IL-10, growth differentiation factor 15 (GDF-15), TNFα, and myeloperoxidase (MPO) were shown to be increased in patients with HFpEF [25,26,27]. These pro-inflammatory cytokines mediate the activation of endothelial cells, which trigger the entire process. Franssen, C. et al. observed high expression of endothelial cell adhesion molecule (VCAM) and E-selectin in coronary microvessels of HFpEF patients (clear signs of endothelial activation) [28]. VCAM attracts circulating monocytes to adhere to endothelial cells, initiating a key step in myocardial inflammation. The monocytes differentiate into macrophages after migrating from the bloodstream into the subendothelial space and then polarize into the M1 or M2 phenotype upon different immunological stimuli. M1 macrophages secrete pro-inflammatory cytokines TNF and IL-1β, which maintain the inflammatory response and myocardial injury; M2 macrophages are involved in fibrosis and diastolic dysfunction by producing pro-fibrotic cytokines TGF-β, IL-10, galectin-3, and osteopontin [29]. The two macrophage phenotypes are often present together during chronic coronary microvascular inflammation [6]. The synergy of M1 (pro-inflammatory phenotype) and M2 (pro-remodeling/anti-inflammatory phenotype) aggravates the feedforward loop of myocardial inflammation and fibrosis. Research shows that the total volume of macrophages doubled in patients with HFpEF compared to non-HFpEF patients [30].

In a systemic pro-inflammatory state, inflammatory cytokines induce the upregulation of ROS-producing NADPH oxidase (NOX), leading to high oxidative stress, increased H_2_O_2_ levels, and endothelial nitric oxide synthase (eNOS) uncoupling [31,32]. H_2_O_2_ levels were significantly elevated in both human and rat samples of HFpEF [28]. eNOS is present in the cardiovascular system and largely produces NO under coupled conditions. However, under uncoupled conditions, eNOS preferentially promotes superoxide production, which exacerbates oxidative stress. It is well known that NO can activate PKG through the NO-cGMP-PKG pathway. Therefore, eNOS decoupling leads to a reduction in NO production and bioavailability, which, in turn, increases the cortical stiffness of the cytoskeleton, which plays a key role in the development of HFpEF ventricular stiffness [33]. In addition, the formation of disulfide bonds (S–S) within Titin caused by ROS is one of the factors that increases myocardial stiffness [34].

### 3.2. Iron Overload Leads to Endothelial Dysfunction

According to the comorbidity–inflammation paradigm, vascular endothelium damage is a key initiating event of myocardial inflammation in HFpEF. Iron overload causes endothelial dysfunction [35], which is thought to be related to excessive ROS [36]. In fact, ROS play physiological roles in signaling cascades at low concentrations, but an increase in ROS production coupled with an insufficient antioxidant response can result in oxidative damage and promote the development of HFpEF. Iron is known to induce an excess of superoxide and hydroxyl radicals via Fenton and Haber–Weis reactions, hydroxyl being the most active and toxic form of ROS [37]. In addition, iron is an important micronutrient for enzyme activity, and the production of ROS by some enzymes, such as NOX and NOS involved in the HFpEF process, and LOX and CYP450 involved in lipid peroxidation, is dependent on the presence of redox-active iron [38,39]. Therefore, we think we have reasons to believe that endothelial dysfunction caused by iron overload is involved in the development of HFpEF.

### 3.3. Macrophage Iron Overload Is Involved in Myocardial Inflammation and Fibrosis

Macrophages resident in myocardial tissue play a central role in iron homeostasis, as they recycle iron by phagocytosing senescent and damaged red blood cells and return most of the iron to circulation [40]. In heart failure, the heme protein released by damaged heart muscle cells increases the source of iron, resulting in macrophage iron overload. On the one hand, iron overload has a marked effect on macrophage polarization. High intracellular iron activates pro-inflammatory M1 macrophages while reducing anti-inflammatory M2 macrophages [41]. This is because, compared to M2 macrophages, M1 macrophages maintain a higher level of intracellular iron and have resistance to drug-induced ferroptosis [42]. According to experiments, iron overload can induce macrophage polarization towards the M1 pro-inflammatory phenotype through the TLR4/TRIF and ROS/acetyl-p53 pathways [43,44]. It further supports that iron overload in macrophages can affect myocardial inflammation, which participates in the development of HFpEF. However, current research on the effect of iron deficiency or excess on macrophages in inflammation is inconsistent, and parts of the studies showed an anti-inflammatory effect of iron [45,46].

On the other hand, cardiac iron-deposition-induced fibrosis has been reported in some iron overload animal models, despite the fact that iron excess reduced M2 macrophages [47,48]. Ishizaka et al. proved that the administration of angiotensin II can induce cardiac fibrosis [49]. Further experiments showed that angiotensin II promotes cardiac fibrosis by inducing iron overload in endothelial cells and macrophages (but not cardiomyocytes) and that iron chelation ameliorated this impact while iron overload exacerbated it [50]. This evidence suggests that iron is involved in the development of angiotensin II-induced cardiac fibrosis. Hypertension is a common comorbidity in patients with HFpEF, and high angiotensin II levels and myocardial fibrosis are present in these patients. This suggests that iron overload may be involved in the development of myocardial fibrosis in HFpEF, and iron chelation may be a potential therapeutic measure.

### 3.4. Cardiomyocyte Iron Overload Impairs Excitation–Contraction Coupling

Systolic cardiac contraction and diastolic relaxation depend on the excitation–contraction coupling (ECC) of cardiomyocytes, which requires proper calcium (Ca^2+^) signaling. The main pathway for iron uptake by the myocardium is dependent on the interaction of Fe^3+^-bound transferrin (TF) with the transferrin receptor 1 (TFR1) (Figure 2). In addition, Fe^2+^ can also enter cells directly through the divalent metal transporter 1 (DMT1) [51]. Serum iron levels have a detrimental effect on these two transporters, resulting in decreased expression of the transferrin receptor to protect cells from iron overload. However, TFR1 and DMT1 are negatively regulated by the serum iron level; this means when serum iron increases, transferrin receptor expression decreases to protect the myocardium from iron overload [51,52]. Recent studies have established that iron can also be transported in excitable cells such as cardiomyocytes by voltage-dependent LTCCs, which are promiscuous divalent transporters [53]. LTCCs are positively regulated by the serum iron level; therefore, they play a dominant role in the transport of myocardial NTBI [53]. Fe^2+^-induced slowing of LTCC current inactivation by Fe^2+^ leads to increased Ca^2+^ entry and overload [54]. In mice investigations, LTCC blockers such as amlodipine and verapamil decreased intracellular myocardial iron buildup and lowered oxidative stress while maintaining diastolic and systolic cardiac function, demonstrating a more solid relationship between LTCCs and myocardial iron deposition [53].

Changes in HFpEF-related Ca^2+^ treatment are currently poorly defined due to the limited availability of cardiac tissue from patients with HFpEF. Nevertheless, there is compelling evidence that impaired calcium homeostasis in patients with HFpEF runs through different periods and stages, from compensatory hypertrophy to clinical symptoms, and is manifested as increased myocardial cell diastolic calcium concentration or abnormal sarcoplasmic reticulum calcium uptake followed by incomplete myocardial relaxation [55,56]. As shown in Figure 2, abnormal calcium homeostasis in cardiomyocytes involves LTCC, RyR2, SERCA2a, and other processes, and these abnormalities may cause diastolic dysfunction alone or in combination, eventually leading to HFpEF. In fact, the Ca^2+^ release of systolic RyR_2_ increased in HFpEF models, and this increased Ca^2+^ release can be attributed to an increased sarcoplasmic reticulum (SR) Ca^2+^ load during Ca^2+^ influx through LTCCs [57,58,59]. Furthermore, impaired SR Ca^2+^ leakage mediated by RyR_2_ and decreased activity of SERCA2a responsible for calcium reuptake was reported in HFpEF rats [60,61].

As a result, increased levels of NTBI may cause cardiac iron overload as well as calcium overload via LTCCs, impairing the ECC of HFpEF cardiomyocytes. Research showed that increased iron levels and LPO mediate ferroptotic cell death, which can also be induced by unregulated calcium levels [62]. Li et al. [63] confirmed that iron treatment increased diabetes-mediated cardiac hypertrophy and calcium homeostasis imbalance, which were correlated with reduced SERCA2a activity and expression. It is worth noting that the Ca^2+^-handling proteins, including RyR2 [64] and SERCA2a [65], are particularly susceptible to oxidative post-translational modifications, which may be important in the context of increased oxidative stress caused by iron overload. A study has shown that the SERCA protein levels are significantly reduced in iron-overloaded rat hearts [66].

### 3.5. Relationship between LPO and HFpEF

Obesity and metabolic syndrome often coincide with disturbances in lipid metabolism. The occurrence and development of HFpEF, therefore, is closely related to LPO according to the comorbidity–inflammation paradigm. It was previously thought that decreased nitric oxide (NO) availability contributed to the pathophysiology of HFpEF [67], but recent research suggests the opposite, for high NO levels caused by iNOS upregulation and NO-induced nitrosative stress are critical components in the pathophysiology of HFpEF [31,68]. It should be noted that ONOO^−^, which is formed by the rapid reaction of NO with superoxide (O_2_^−^) radicals, is a powerful inducer of LPO. Excess NO-induced accumulation of lipid peroxides promotes ferroptosis. Further study revealed a high NO-driven dysregulation of the Xbp1s–FoxO1 axis as a pivotal mechanism in the pathogenesis of cardiometabolic HFpEF, FoxO1 depletion, as well as over-expression of the Xbp1s (a spliced form of the X-box-binding protein 1) arm of the UPR (unfolded protein response) in cardiomyocytes; each ameliorates the HFpEF phenotype (diastolic dysfunction and exercise tolerance) in mice and reduces myocardial lipid accumulation [69]. Excitingly, it has preliminarily been shown that GPX4, a key molecule that regulates ferroptosis, is reduced in HFpEF mice, confirming the presence of ferroptosis in HFpEF, and after treatment with the hypoglycemic drug imeglimin, GPX4 is restored as are systemic glucose metabolism, visceral obesity, and cardiac abnormalities [70]. In addition, elevated levels of H_2_O_2_ and LPO end product malondialdehyde (MDA) were observed in obesity-related HFpEF rats [71], with a clinical study showing that the degree of MDA elevation is related to heart function and the course of heart failure [72]. It has been suggested that another LPO end product, 4-hydroxynonenal (4-HNE), an important marker of ferroptosis, may play a particularly sinister role in metabolic syndrome and related disorders [73]. Furthermore, 4-HNE can cause cardiac hypertrophy by inhibiting mitochondrial energy by regulating enzyme NADP+-dependent isocitrate dehydrogenase [74], further suggesting that lipid peroxidation may play an important role in the pathological process of HFpEF.

## 4. Research Progress of Anti-Ferroptosis in HFpEF

In recent years, the role of ferroptosis in cardiovascular disease has received increasing attention. Cells undergoing ferroptotic cell death are characterized by the accumulation of lipid peroxide, metabolism of PUFA, excess iron, and cannot be rescued by inhibitors of apoptosis or other cell death processes [75]. Inflammation, oxidative stress, and myocardial injury induced by these biological processes are inextricably linked to HFpEF. Targeting ferroptosis may be a potentially effective treatment for HFpEF.

Recent research found that GPX4 was reduced in HFpEF mice with obesity and impaired glucose tolerance, and imeglimin treatment not only improved their abnormal systemic glucose metabolism and visceral obesity, but also suppressed the upregulation of iNOS and restored expression of Xbp1s and GPX4 [70]. In addition, the expression of GPX4 and ferritin (Fth1) was significantly downregulated in a pressure overload-induced heart failure model in rats, while knockdown of TLR4 or NOX4 gene restored the expression of GPX4 and Fth1 in cardiomyocytes and improved ventricular remodeling and LV function in rats with heart failure [76]. It is worth noting that Sai Ma et al. [77] demonstrated that ferroptosis might be a key mechanism in a rat model of HFpEF with proteomics, which was blocked following treatment with canagliflozin, a sodium–glucose cotransporter 2 inhibitor (SGLT2i).

Alpha-lipoic acid (ALA) is an ideal antioxidant that scavenges ROS, chelates divalent transition metals (including Fe^2+^), and regenerates some endogenous antioxidants such as glutathione. Discontinuous treatment with ALA in HFpEF rats prevented the development of cardiac hypertrophy and both attenuated and delayed the onset of diastolic dysfunction through anti-obesity effects, anti-inflammatory effects, and reducing lipid metabolism disorders (raising the GSH/GSSG ratio and lowering circulating MDA levels) [71]. As the only drug shown to improve the prognosis of HFpEF to date, acute treatment with empagliflozin reduced inflammation (i.e., ICAM-1, VCAM-1, TNF-α, and accumulation of IL-6) and oxidative stress (H_2_O_2_, 3-nitrotyrosine, glutathione, and LPO) in the cytosol and mitochondria of cardiomyocytes in mice and humans with HFpEF, and thus improved endothelial and myocardial function [78,79,80]. These findings suggest that anti-ferroptosis in LPO may be an effective measure for the treatment of HFpEF to some extent. In addition, CoQ10 supplementation can inhibit LPO and ferroptosis through the FSP1–CoQ10–NADPH pathway. However, current studies on CoQ10 improving myocardial diastolic function have shown conflicting results. Adarsh et al. [81] showed that CoQ10 significantly improved exercise tolerance and diastolic dysfunction in patients with hypertrophic cardiomyopathy, but no positive results were shown in two other studies [82,83]. This may be due to differences in the diastolic function assessment criteria and biologic agents (ubiquinone and ubiquinone alcohol) that led to changes in bioavailability [84]; thus, the effectiveness of CoQ10 needs further confirmation.

Lipocalin (APN) is an adipokine that can regulate cardiac remodeling by interacting with multiple intracellular signaling pathways [85]. Recently, low lipocalin levels were found to increase the propensity for diastolic heart failure and diastolic dysfunction in an experimental rat model [86]. Essick et al. [87] further showed that APN improves cardiomyocyte autophagy by inhibiting H_2_O_2_-induced AMPK/mTOR/ERK-dependent mechanisms, demonstrating its antioxidant therapeutic potential against HFpEF. Among them, AMP-dependent protein kinase (AMPK) and rapamycin (mTOR) can inhibit ferroptosis through different pathways. Energy stress activates AMPK, which suppresses PUFA and other fatty acid production, at least in part by phosphorylating and inactivating acetyl coenzyme A carboxylase (ACC), inhibiting ferroptosis [88]. Bayeva et al. [89] discovered that mTOR inhibited mRNA binding protein (TTP), reducing its binding and boosting TfR1 mRNA degradation in cardiomyocytes. TTP knockout mice had impaired cardiac function and iron deficiency [90], which strongly corroborate this mechanism.

According to the current paradigm of HFpEF development, coronary microvascular endothelial inflammation and dysfunction is the initial and critical mechanism for cardiac remodeling. Hypertensive cardiac remodeling is characterized by hypertrophy, fibrosis, and inflammation [91]. Effectively inhibiting pathological myocardial hypertrophy has an irreplaceable role in the prevention and treatment of HFpEF. Zhang et al. confirmed that cardiac microvascular endothelial cell ferroptosis exacerbated Ang II-induced cardiomyocyte hypertrophy and oxidative stress [92]. The cystine/glutamate antiporter SLC7A11 (also commonly known as xCT) functions to import cystine for glutathione biosynthesis and antioxidant defense. Recent studies have shown that xCT can inhibit Ang II-mediated cardiac hypertrophy by blocking ferroptosis. Thus, upregulation of xCT may represent a new therapeutic approach for hypertrophic cardiac diseases [93].

## 5. Conclusions

Appropriate treatment of HFpEF is considered one of the largest unmet needs in the cardiovascular field because of its high prevalence, adverse outcomes, and lack of approved beneficial therapies [3,94]. Recently, the EMPEROR-Preserved study for HFpEF patients reported that empagliflozin treatment results in a significant reduction in the occurrence of the main composite endpoints for this condition [80]; because of this, SGLT2i, represented by empagliflozin, has obtained a level 2a recommendation for HFpEF treatment in the 2022 AHA/ACC/HFSA guidelines [95]. This is a major milestone in the treatment of HFpEF, and so SGLT2i is of significant interest. Canagliflozin was shown to block ferroptosis and treat HFpEF [77], which demonstrated that the intervention of ferroptosis could be used as a new attempt. Further research is needed to clarify the causal relationship and specific mechanism between ferroptosis and HFpEF and provide an opportunity for designing new therapeutic interventions.

Clinical studies showed that Fth1 was positively correlated with amino-terminal pro-B-type natriuretic peptide (NT-proBNP) and the ratio of mitral peak velocity of early filling to early diastolic mitral annular velocity (E/e’) [96,97,98]. Additionally, ROS generated by the Fenton reaction can reduce NO bioavailability in cardiomyocytes and promote myocardial hypertrophy and ventricular wall stiffness through the NO–cGMP–PKG pathway [33], suggesting that iron overload was involved in LV diastolic dysfunction in HFpEF patients. Thus, early detection of myocardial iron overload is important. Many researchers have reported that cardiovascular magnetic resonance imaging (cMRI) with T2* < 20 msec remains, at present, a more reliable assessment of iron overload in the heart [99,100,101]. However, it should be noted that iron deficiency is also one of the common comorbidities in heart failure, both in HFrEF and HFpEF [102]. The 2021 ESC guidelines for the diagnosis and treatment of acute and chronic heart failure point out that intravenous iron supplementation should be considered in symptomatic HFrEF patients recently hospitalized for HF and iron deficiency, defined as serum ferritin < 100 ng/mL or serum ferritin 100–299 ng/mL with TSAT < 20%, to reduce the risk of HF hospitalization [103], but evidence on iron supplementation in HFpEF is still missing. The safety of iron supplementation is critical because of the iron sensitivity of cardiomyocytes. In general, secondary iron overload is often caused by blood transfusions, long-term iron supplementation, etc. Therefore, attention should be paid to the calculation of the dose of iron supplementation and the detection of iron metabolism indicators during iron supplementation treatment to avoid iron overload. However, many patients develop iron overload before starting iron supplementation therapy, and increased iron absorption due to ineffective erythropoiesis is now considered to be another major cause of iron overload [104]. Iron uptake, storage, and consumption of cardiomyocytes involve a series of complex and delicate regulatory mechanisms. Interestingly, numerous studies have shown that imbalances in iron metabolism, including both iron deficiency and iron overload, can cause myocardial cell damage through a variety of complex processes, affecting structure and function [105,106,107]. Walter et al. [108] found a U-shaped relationship between stored iron and the incidence of heart failure in mice. Whether iron supplementation or iron chelate therapy is beneficial to patients with HFpEF remains unclear. The exact mechanism between iron content, ferroptosis, and cardiomyocyte injury needs to be further investigated.

The regulatory mechanisms of myocardial cell death are of great research value and transformational significance for heart failure because the myocardium is non-renewable. Ferroptosis and its regulatory processes are closely related to the development of HFpEF and deserve further attention. Second, the incidence of ferroptosis includes the expression and control of multiple genes, its signaling pathways are complicated, and the processes have not been fully revealed. More in-depth investigations are expected to uncover the currently unknown molecular processes of ferroptosis and provide more scientific evidence for targeting ferroptosis in HFpEF.

## Figures and Tables

**Figure 1 cells-11-02842-f001:**
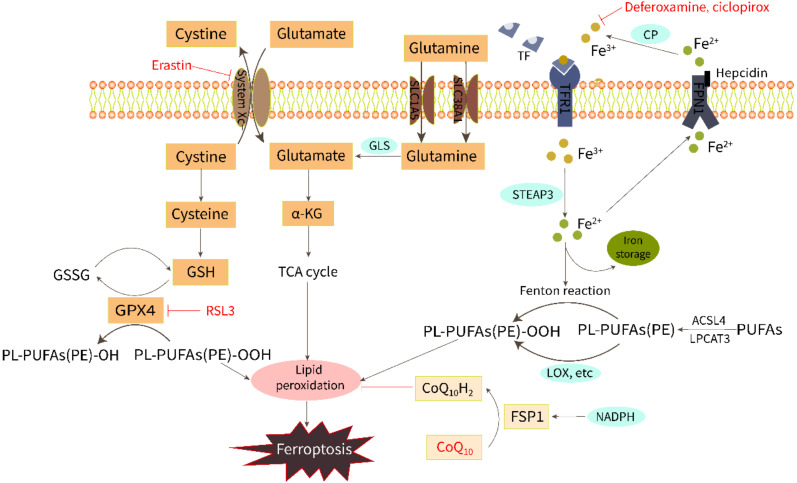
Regulation of ferroptosis by the metabolism of iron, lipids, and amino acids.

**Figure 2 cells-11-02842-f002:**
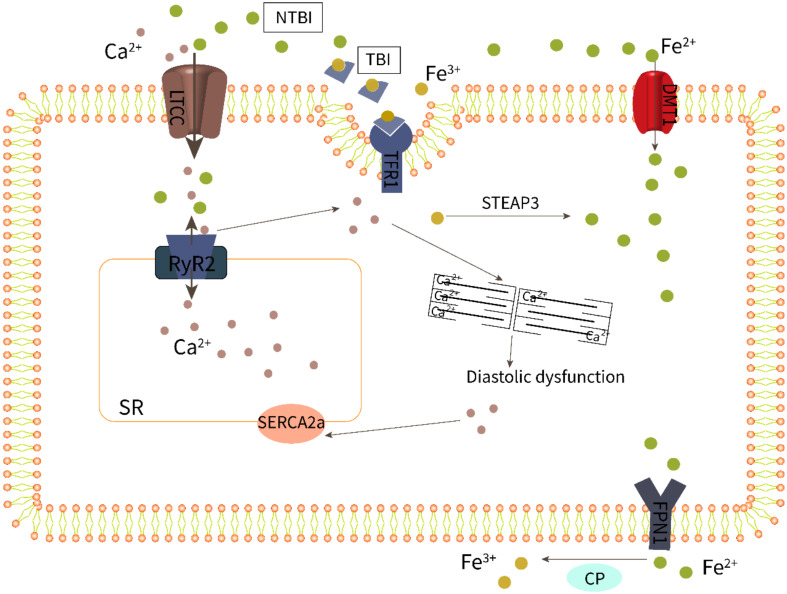
Iron overload impairs excitation–contraction coupling of cardiomyocytes.

## Data Availability

Not applicable.

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
