# Peer review of "Ferroptosis: The Potential Target in Heart Failure with Preserved Ejection Fraction"

_cells, 2022, doi:10.3390/cells11182842_

Round 1

Reviewer 1 Report

The authors reviewed the ferroptosis and its regulatory processes in HFpEF. The regulatory mechanisms of myocardial cell death are of great research value and transformational significance for heart failure because myocardium is non-renewable.  This review is very interesting to the reviewer, but need to be corrected in several important respects.

1) Ferroptosis is a form of cell death, here, its means myocardial cell death. Therefore, Ferroptosis leads to myocardial fibrosis, and then reduced contractility ( reduced EF) , and is it not associated with REF rather than PEF?

How do you explain the connection between myocardial cell death and cardiac hypertrophy?

2) They mixed up with cardiac hypertrophy, cardiac remodeling, and cardiac dysfunction, but each has a different meaning, so it is better to use the correct terminology.

3) Future Directions and Conclusion section

In this section, the definition of iron overload is unclear.

The iron deficiency and iron excess that cause anemia are not clearly defined, and it is unclear whether iron administration is good or bad in PEF correlated with Ferroptosis.

Reviewer 2 Report

The authors present a really beautiful review on a complex topic: defining molecular mechanisms for chronic disease symptoms is not an easy challenge. The structure of the review, covering essential biochemistry and immunological background, is really excellent. While most of the English is fine, there are a several sentences which need to be corrected, e.g.:

line 65 "Excess Fe2+ can activate(...) by catalyzing oxygenase to induce ferroptosis" do you mean catalyzing oxygenase activity?

line 251 "Excitingly, it has been preliminarily shown that GPX4, a key molecule that mediates iron death..." GPX4 is a key molecule, but the other way round, it inhibits ferroptosis.

line 331 "HFpEF is considered one of the largest unmet needs" do you mean "appropriate treatment of HFpEF..."?

As a conclusion, I see a fantastic review covering the most important facts known about disease mechanisms potentially resulting in heart failure and highlighting potential therapeutic options resulting from those insights. After careful correction of some maybe poorly translated sentences I would happily endorse publication of this manuscript.

Round 2

Reviewer 1 Report

good job